# Effects of Phenolics on the Physicochemical and Structural Properties of Collagen Hydrogel

**DOI:** 10.3390/polym15244647

**Published:** 2023-12-08

**Authors:** Sadia Munir, Wei Yue, Jinling Li, Xiaoyue Yu, Tianhao Ying, Ru Liu, Juan You, Shanbai Xiong, Yang Hu

**Affiliations:** 1College of Food Science and Technology, Huazhong Agricultural University, Wuhan 430070, China; bajwa.uos@gmail.com (S.M.); xiaoyueyu@webmail.hzau.edu.cn (X.Y.); yingtianhao@webmail.hzau.edu.cn (T.Y.); liuru@mail.hzau.edu.cn (R.L.); juanyou@mail.hzau.edu.cn (J.Y.); xiongsb@mail.hzau.edu.cn (S.X.); 2Bioactive Peptide Technology Hubei Engineering Research Center, Jingzhou 434000, China

**Keywords:** collagen, gallic acid, ellagic acid, hydrogels, storage modules

## Abstract

In the current era, the treatment of collagen hydrogels with natural phenolics for the improvement in physicochemical properties has been the subject of considerable attention. The present research aimed to fabricate collagen hydrogels cross-linked with gallic acid (GA) and ellagic acid (EA) at different concentrations depending on the collagen dry weight. The structural, enzymatic, thermal, morphological, and physical properties of the native collagen hydrogels were compared with those of the GA/EA cross-linked hydrogels. XRD and FTIR spectroscopic analyses confirmed the structural stability and reliability of the collagen after treatment with either GA or EA. The cross-linking also significantly contributed to the improvement in the storage modulus, of 435 Pa for 100% GA cross-linked hydrogels. The thermal stability was improved, as the highest residual weight of 43.8% was obtained for the hydrogels cross-linked with 50% GA in comparison with all the other hydrogels. The hydrogels immersed in 30%, 50%, and 100% concentrations of GA also showed improved swelling behavior and porosity, and the highest resistance to type 1 collagenase (76.56%), was obtained for 50% GA cross-linked collagen hydrogels. Moreover, GA 100% and EA 100% obtained the highest denaturation temperatures (Td) of 74.96 °C and 75.78 °C, respectively. In addition, SEM analysis was also carried out to check the surface morphology of the pristine collagen hydrogels and the cross-linked collagen hydrogels. The result showed that the hydrogels cross-linked with GA/EA were denser and more compact. However, the improved physicochemical properties were probably due to the formation of hydrogen bonds between the phenolic hydroxyl groups of GA and EA and the nitrogen atoms of the collagen backbone. The presence of inter- and intramolecular cross-links between collagen and GA or EA components and an increased density of intermolecular bonds suggest potential hydrogen bonding or hydrophobic interactions. Overall, the present study paves the way for further investigations in the field by providing valuable insights into the GA/EA interaction with collagen molecules.

## 1. Introduction

In the present era, hydrogels are an attractive class of materials. Hydrogels have been known for several years as appealing scaffolds because have they highly expanded and interconnected structural networks, which give them the ability to encapsulate the bioactive compounds and effectively transfer mass [1]. Recently, more and more researchers have paid attention to natural polymer hydrogels due to their high safety level, low immunogenicity, good biocompatibility, biodegradability, hydrophilic nature, and abundant availability [1,2,3], including proteins. Collagen is abundantly found in human tissues and is particularly abundant in load-bearing structural systems such as bones, skin, lungs, and tendons. In addition, collagen can also improve cellular adherence and promote extracellular matrix production in proliferating cells [4]. Furthermore, collagen molecules have many reactive groups that can be used to modify the collagen [5] and also function in numerous organic natural body functions such as tissue regeneration, healing, control of tissue-related diseases, cellular response, and structuring [6]. In addition, collagen suffers from the limitations of relatively weak mechanical properties, chemical stability, and resistance to enzymatic degradation [5], although it is widely used as an ingredient to improve the consistency, elasticity, and stability of food products.

Plant-based phenolic composites are considered to be the most vital bioactive composites. These compounds contain a number of hydroxyl groups and have diverse biological functions such as structural support, pigmentation, chemical defense, and radiation prevention [7,8]. Gallic acid (GA) is also a phenolic compound of plant origin and is known as (3,4,5-trihydroxybenzoic acid). Moreover, GA has been shown to have a variety of properties in biomedical studies, including anti-allergic, anti-fungal, anti-inflammatory, anti-cancer, anti-viral, anti-mutagenic, and anti-carcinogenic properties [7,9,10]. In addition, from a medical point of view, gallic acid plays an important role in the protective mechanism against reactive oxygen species and free radicals. It breaks the free radical chains through hydroxyl groups [6,11]. Furthermore, GA was reported by Thanyacharoen et al. to be a bioactive and stable agent in chitosan/PVA-based hydrogels [12]. It was also reported by Jiang et al. [13] that GA can increase the release efficiency of chitin-based hydrogels.

Alternatively, ellagic acid (EA), a representative of flavonoids found in a variety of fruits such as pomegranate, pecans, and berries, has received extensive attention due to its numerous antioxidant, cytotoxic, radical-scavenging, anti-viral, anti-inflammatory, anti-carcinogenic, and anti-apoptotic properties [14,15]. Ellagic acid is a dilactone of hexahydrooxydiphenic acid. It is usually produced by plants through the hydrolysis of tannins such as ellagitannins [16]. It contains four hydroxyl groups and these hydroxyl groups can increase the antioxidative action of lipid peroxidation to protect the cell from oxidative destruction [14]. Therefore, EA has hydrophilic characteristics due to its structural appearance, which includes a planar biphenyl, and a lipophilic component connected by two lactone rings, and four hydroxyl groups. These hydroxyl groups combine with the lactone groups to form a hydrophilic unit [17]. The hydrophilic region of the EA molecule plays an important role in its biological activity. Notably, this is due to the presence of both hydrogen-bonding acceptor (lactone) and donor (-OH) sites. In particular, the phenolic hydroxyl groups in EA can be separated under physiological conditions [18,19]. In addition, it has been reported by Huang et al. that EA can cross-link with PEG-based hydrogels and reduce the viability of human oral cancer cells [20].

The primary purpose of this study was to prepare collagen hydrogels with improved physicochemical properties via the cross-linking of natural phenolic compounds. Gallic acid or ellagic acid were added as potential cross-linkers. In particular, the influences of phenolic compounds on the physicochemical properties, such as the thermal, structural, enzymatic, and morphological properties of collagen-based hydrogels, were examined. In previous studies, gallic acid and ellagic acid have never been prepared in collagen-based hydrogels and compared to each other. In addition, XRD, FTIR, water retention, enzyme degradation, porosity, and swelling ratio were investigated. Finally, the SEM examination of the cross-linked hydrogels was also carried out. This study could contribute to providing a new vision in the biomaterials or biomedical industries based on the cross-linking of collagen hydrogels with natural phenolic compounds such as gallic acid or ellagic acid.

## 2. Material and Methods

### 2.1. Materials

Collagen extraction was performed using fresh grass carp (*Ctenopharyngodon idella*) obtained from a local slaughterhouse. Following the procedure described by Zhu et al. [21], the extraction method used was a combination of acid and pepsin extraction. The pepsin enzyme used in the extraction process was purchased from Bio-Sharp Company. Gallic acid and ellagic acid used in the study were purchased from Shanghai Aladdin Bio-Chem Technology Co., Ltd., Shanghai, China. Lyophilized type I collagenase isolated from *Clostridium histolyticum* (freeze-dried powder, ≥125 CDU/mg solid) was purchased from Sigma, St. Louis, MO, USA. All other reagents used in the experiments were of high analytical grade.

### 2.2. Production of Collagen

Using a modified approach, collagen was isolated from the skin of freshly bought grass carp *(Ctenopharyngodon idella)* at 4 °C [21]. To summarize, the skin was washed, sliced into small pieces, and soaked for 72 h in 0.01 M NaOH (1:20, *w*/*v*). To remove non-collagenous constituents, the solution was changed after every 8 h. The skin pieces were then immersed in 10% (*w*/*w*) isopropanol for 24 h to remove fat, and then neutralized with double-distilled water. Based on the dry weight of the skin, the pieces were combined with 0.5 M acetic acid (1:50, *w*/*v*) and 2% pepsin (1:3000; Sigma, USA). The mixture was stirred at 4 °C for 2 days. To separate the pepsin-soluble collagen solution, it was filtered and centrifuged (10,000 rpm, 10 min). For purification, 1.5 M (NH_4_)_2_SO_4_ was added overnight, followed by 0.5 M acetic acid. The resulting purified collagen was resolubilized in 0.5 M acetic acid and then dialyzed with 0.04 M disodium dihydrogen pyrophosphate solution for 2 days, 0.02 M disodium dihydrogen pyrophosphate solution for 3 days, and finally with double-distilled water for 2 days. The collagen solution was lyophilized and kept in a dark and dry place for future studies. This ensured the high quality of the collagen from the grass carp skin.

### 2.3. Fabrication of Cross-Linked Collagen Hydrogels

Pure collagen hydrogels were prepared in accordance with the method of Zhu et al. [21], with slight modifications as illustrated in Figure 1. Freeze-dried collagen was reconstituted by dissolving the collagen in 0.5 M acetic acid. The ratio was (10 mg:1 mL) 10 parts of collagen to 1 part acetic acid, and the process was performed at 4 °C. The solution of collagen and acetic acid was subjected to gentle stirring until the complete dissolution of the collagen was achieved. The pH of the collagen solution was adjusted to a neutral range between pH 7.0 and 7.5 and it was adjusted using either 2 M NaOH or 2 M acetic acid with continuous monitoring and necessary adjustments to ensure the desired pH. Next, 5 mL of the neutralized collagen solution was added to individual 24-well plates. These plates were then incubated at 37 °C for 4 h. This incubation time allowed the collagen to modify as a gel. The hydrogels were stored at 4 °C overnight after the initial incubation. Further gelation and overall stabilization of the collagen hydrogels were promoted by this additional step.

For the preparation of cross-linked collagen hydrogels, GA and EA were dissolved individually in tri-distilled water at different concentrations (0%, 1%, 5%, 10%, 30%, 50%, 100%, and 200% *w*/*w* based on the dry weight of the collagen hydrogels) at a pH range of 4.5–5.5. The solutions containing GA and EA were gently shaken at room temperature for approximately 30 min. The collagen hydrogels were then immersed in these prepared solutions. The immersion was performed at room temperature, approximately 25 °C, for a period of 4 h. The cross-linked collagen hydrogels were carefully washed with distilled water to remove any unbound GA and EA after the 4 h immersion. Finally, the gels were stored at 4 °C for further processing.

### 2.4. Collagen Hydrogel Characterization

#### 2.4.1. Dehydration of Hydrogels

Prior to lyophilization, all hydrogels were placed at −80 °C for complete freezing. Dehydration of the frozen hydrogels was performed in a lyophilizer for a minimum of 24 h to remove all the moisture. Lyophilized hydrogels were subjected to thermogravimetric curve, DSC, SEM, porosity, water-holding capacity, swelling behavior, FTIR, and XRD characterization to verify the water-holding capacity of the network in the structure after lyophilization.

#### 2.4.2. Determination of Porosity

The determination of the porosity of the hydrogel was in accordance with Zhu et al. [21]. First, a 25 mL beaker was filled with ethanol, and its weight was *m*_1_. The lyophilized sample (*m*_0_) was then immersed into the beaker, which was then ultrasonically degassed to permeate the hydrogel with ethanol, and the total weight was calculated as *m*_2_. The ethanol was then carefully scraped from the hydrogel’s surface, and the weight of the left half was measured as *m*_3_. Following that, the hydrogel’s porosity was then estimated using the following formula:Porosity (%) = (*m*_2_ − *m*_0_ − *m*_3_)/(*m*_1_ − *m*_3_) × 100(1)

#### 2.4.3. Swelling Ratio and Water-Holding Capability

The swelling ratio was quantified according to Zhu et al. [21] at 25 °C by soaking the weighed (W_0_) freeze-dried cylindrical specimens in phosphate-buffered saline (0.1 M PBS, pH 7.4). After soaking for 1, 3, 5, 8, 12, 20, 50, and 90 min, the swollen hydrogels were removed. After the removal of excess water, the hydrogels were immediately weighed (W_t_). The following formula can be used to calculate the swelling ratio (SR):SR = (W_t_ − W_0_)/W_0_(2)

The following method was used to determine the water-holding capacity (WHC). Briefly, the freshly prepared samples (equilibrium swollen hydrogel in water) were first weighed after the water had been wiped off from the surface with filter paper (W_swollen_). Subsequently, the samples were then lyophilized and weighed as W_lyophilized_. The following formula can be used to calculate the percentage of WHC:WHC (%) = (W_swollen_ − W_lyophilized_)/W_swollen_ × 100(3)

#### 2.4.4. Thermogravimetric Analysis (TGA) of Collagen Hydrogels

Thermogravimetric (TG) curves of collagen hydrogels were obtained using a TGA-2050 instrument (Mettler-Toledo). The instrument was operated under a nitrogen atmosphere and heated between 30 °C–550 °C at a heating rate of 10 °C/min. The rehydrated hydrogels were used for the TGA test with a sample size of approximately 5 mg.

#### 2.4.5. Thermal Transition Analysis of Collagen Hydrogels

The thermal transition temperature of the collagen hydrogels was studied using differential scanning calorimetry (DSC) (DSC 200PC, Netzsch, Germany. The specimens (5 mg) were precisely weighed and placed in sealed aluminum dishes. At a heating rate of 5 °C/min over a temperature range of 25 °C–110 °C, the aluminum pans were scanned under a nitrogen atmosphere. The empty, sealed aluminum pan, which was sealed, was used as the reference point. The temperature of the endothermic peak was reported as the temperature of denaturation (Td).

#### 2.4.6. Analysis by X-Ray Diffraction (XRD)

The X-ray diffraction structures of the collagen hydrogels were analyzed using CuKa radiation from a rotating anode generator operated at 40 kV and 40 mA in the 2θ range (4–60 °C) with a mono-single filter at a scan rate of 10 °C min^−1^ (D8 Advance, Bruker, Germany).

#### 2.4.7. Fourier Transform Infrared Spectroscopy (FTIR) Analysis

On a germanium, single crystal FTIR spectrophotometer, the infrared spectra of all lyophilized gel powders were obtained from tablets containing (0.8–1 mg) collagen hydrogel in approximately 100 mg potassium bromide (KBr). All freeze-dried gel powders’ IR spectra were obtained from tablets containing (0.8–1 mg) collagen hydrogel in 100 mg potassium bromide (KBr). The tablets were placed on the single-reflection germanium crystal cell using an FTIR spectrophotometer. Signals from 4000 to 400 cm^−1^ were acquired for 64 scans at a data acquisition rate of 4 cm^−1^ per point and compared to a background spectrum collected from a clean, empty cell.

#### 2.4.8. Measurements of Dynamic Rheology

The hydrogels (diameter = 20 mm, thickness = 5 mm) prepared from self-assembled neutral collagen solutions (10 mg/mL) and incubated for 2 h were subjected to a dynamic time sweep for 60 min (37 °C) at the different frequencies (1–10 Hz) to monitor the gelling behavior. Furthermore, the dynamic temperature was also set at 37 °C. During the test, a rheometer (AR2000ex, TA, Woodland, CA, USA), with a parallel stainless-steel plate (diameter = 40 mm, gap = 1 mm) was used to determine the storage and loss modulus values. The frequency sweep was adjusted to 0.01–10 Hz. A deformation of 2% was selected for all samples. The temperature was controlled by using a Peltier temperature controller. A solvent trap was used to prevent water loss from the samples during the measurement. The value of the tangent δ (tan δ) was also calculated as the G′′/G′ ratio, which reflects the thermal energy loss.

#### 2.4.9. Enzymatic Analysis of Stability

The in vitro enzymatic degradation of collagen-based hydrogels was carried out with the use of type I collagenase that was derived from Clostridium. Lyophilized hydrogels were first swollen by immersion in phosphate-buffered saline (PBS) at a pH of 7.0 until complete swelling was achieved. Each hydrogel was then placed in 1 mL enzyme hydrolysate containing type I collagenase (200U, Sigma) and 0.01 M CaCl_2_ and incubated at 37 °C for 24 h. 0.2 mL of 0.25 M EDTA was added to the hydrogel mixture followed by cooling in an ice bath to complete the degradation process. The mixture was then centrifuged at 5000× *g* for 10 min at 4 °C. A quantity of 2 mL of the resulting supernatant was collected. Quantities of 1 mL of chloramine T and perchloric acid were added to the supernatant. Each was allowed to stand for 20 min and 5 min, respectively. Then, 1 mL of paradimethylaminobenzaldehyde (DMAB) was added. The mixture was incubated in a water bath at 60 °C for 20 min. Ultraviolet spectroscopy at 560 nm was used to quantify the hydroxyproline content. The conversion coefficient between collagen and hydroxyproline in aquatic animals was calculated to assess the degree of hydrogel biodegradation. Specifically, in the absence of GA or EA, the percentage of hydroxyproline released from the collagen-based hydrogel was compared to that of fully degraded collagen. For reference purposes, untreated collagen was used as a control.

#### 2.4.10. Scanning Electron Microscopy (SEM)

Collagen hydrogels cross-linked with GA or EA for morphological characterization were prepared as described by Liu et al. [22] with slight modifications. Briefly, freshly prepared hydrogels were cut into fragments (~2 × 2 mm) and then immersed in 2.5% glutaraldehyde in 0.2 M phosphate (pH 7.2) overnight and dehydrated in graded ethanol solution with a series of concentrations (30%, 50%, 70%, 80%, 95%, and 100%). Subsequently, the samples were then treated with isoamyl acetate for a period of 15 min. The hydrogels were then freeze-dried with the use of a lyophilizer. After that, the hydrogels were coated with a layer of gold. An ultrahigh-resolution field emission scanning electron microscope was used to observe the microstructure, and SEM images were observed using a JSM-5610 SEM (JEOL, Tokyo, Japan), with 20 kV acceleration voltage. The magnification was 3 K, 8 K, and 15 K times.

#### 2.4.11. Statistical Evaluation

Prism and Origin software versions 8.5 (SAS Institute Inc., Cary, NC, USA) were used for data analysis. To detect statistically significant differences, Duncan’s multiple range test was performed. The level of significance was fixed at *p* < 0.05. By finding significant differences across experimental groups, this statistical technique provides compelling evidence for the reported results.

## 3. Results and Discussion

### 3.1. Porosity Measurements

The porosity, an important factor for collagen hydrogels, is illustrated in Figure 2A. In general, higher porosities have been studied, particularly for collagen-based hydrogels. These were thought to be valuable for supporting nutrition and cell metabolism during cell proliferation and adhesion in tissues [23]. The control hydrogels exhibited a lower porosity level of ~87.51%, and a similar tendency was observed in GA 1%, EA 1%, GA 200%, and EA 100% (~88.96%, 88.21%, 89.83%, and 89.51%, respectively). Notably a concentration-dependent pattern was developed in the porosity of the collagen cross-linked hydrogel, which increased with the increasing concentrations of GA but decreased with the GA 200%, which might be due to the higher concentration of GA. Conversely, the porosity of EA cross-linked hydrogels presented an increase with increasing EA concentrations up to 30%, followed by a decrease with higher EA concentrations. GA 50% exhibited the highest porosity (98.24%) among all hydrogels containing GA and EA at all concentrations. Gallic acid exhibited significantly superior porosity than ellagic acid in the cross-linking of collagen hydrogels according to the findings of the present study. The distinction in porosity can be attributed to the differences in chemical structures and the capacity of GA and EA to bind with collagen. At higher concentrations, the porosity of GA/EA cross-linked hydrogels decreased as GA and EA bound to collagen through hydrogen and hydroxyl bonds [24]. In particular, higher concentrations of phenolic acid appeared to disrupt the self-assembly of collagen, resulting in heterogeneous pore topologies with lower porosity at 200% of GA and 50%, 100%, and 200% of EA. Remarkably, the porosity of collagen hydrogels cross-linked with GA/EA demonstrated dynamic behavior. It was characterized by an initial increase followed by a subsequent reduction in the porosity. The initial porosity rise can be attributed to the introduction of cross-linking agents into the hydrogels [25]. GA or EA facilitated the formation of interconnected pores within the hydrogel matrix. The subsequent reduction in porosity with an increasing concentration of GA/EA cross-linked hydrogels decreased because of matrix enhancement, potentially driven by stronger intermolecular interactions and structural rearrangements within the hydrogel network. Prior studies have also reported that higher porosity is correlated with improved cell viability, tissue formation, wound healing, cell proliferation, and cell permeation [21,25].

### 3.2. Water-Holding Capability

Numerous critical elements have contributed to the observed phenomenon in the context of water-holding capacity. Hence, immersion of GA and EA within the collagen matrix boosted the hydrogel’s hydrophilicity, aggregating both water in absorption and water retention [26,27]. Furthermore, the water-holding capacity (WHC) of GA/EA cross-linked hydrogels presented concentration-dependent descriptions, as shown in Figure 2B. There was an increase in WHC at lower concentrations of GA or EA cross-linked hydrogels. A number of assorted factors support these phenomena. Furthermore, the immersion of GA and EA inside the hydrogel structure may have provided additional water molecule binding sites, contributing to an increase in WHC in cross-linked collagen hydrogels [28]. However, when the concentration of GA/EA in cross-linked hydrogels exceeded a predefined limit, WHC decreased. The reduction was due to the increasing density and compactness of the hydrogel matrix at higher cross-linking concentrations. Increased concentrations of GA or EA result in an extraordinary production of binding sites within the collagen network, resulting in a denser matrix that inhibits water entry and retention. When the WHC of GA and EA hydrogels at equal concentrations was compared, it was clear that GA hydrogels had better WHC compared to EA. The difference can be attributed to the different chemical structures and cross-linking abilities of GA and EA. Gallic acid has a higher tendency for water molecule binding and a more dynamic interaction with collagen, resulting in increased water absorption and retention [29]. On the other hand, the WHC of collagen hydrogels cross-linked with EA increased with increasing EA content up to a concentration of 30%. Then, the WHC gradually decreased. Moreover, the highest WHC was found in collagen hydrogels cross-linked with GA, especially at a 50% GA concentration (89.07%), whereas the lowest WHC was found in collagen hydrogels cross-linked with EA at a 200% EA concentration (64.92%). These results highlight the significant influence of GA and EA cross-linking on WHC, which may have implications for a variety of biomaterial applications. However, Lin et al. [30] conducted research that also supported the idea that improved WHC in cross-linked collagen hydrogels is beneficial for biomedical engineering in biomaterials.

### 3.3. Swelling Property

The swelling property is also known as a vital property of hydrogels and is usually related to the moisture transfer across the hydrogels or the retention of moisture within the system from the environment. The swelling ratio (SR) was primarily affected by external solution variables such as charge number and ionic strength, as well as polymer features such as network flexibility, the presence of hydrophilic functional groups, and the degree of cross-linking density [31]. The effects of cross-linking on the swelling properties of GA/EA cross-linked hydrogels are shown inFigure 2C,D. The swelling ratio in the neutral solution (native collagen) was lower than that in the acidic solution (GA/EA cross-linked hydrogels). All hydrogels presented swelling within the first 20 min after immersion in the solution. The swelling ratio improved marginally from 8 to 50 min, then remained nearly constant from 50 to 90 min, indicating that the swelling equilibrium had been reached. The GA/EA addition enhanced the swelling ratio of GA/EA cross-linked collagen hydrogels. Hydrogels quickly absorb aqueous solutions in acidic circumstances, and the hydroxyl and amino groups in the hydrogels become highly protonated [32]. However, the immersion of EA promoted the swelling ratio when the EA was less than 100% based on the dry weight of collagen. The formation of hydrogen bonds between the phenolic hydroxyl groups of GA and EA and the nitrogen atoms of the collagen backbone is most likely responsible for the enhanced swelling ratio. On the other hand, the swelling ratio of EA cross-linked hydrogels with higher EA concentrations (100% and 200%) was lower than that of pure collagen hydrogel. However, this was due to the significant self-polymerization of EA, which inhibits the extension of collagen fibrils in PBS solution to achieve a reduced swelling ratio [33,34]. However, the swelling ratio was remarkably increased (about 3–15% (*p* < 0.05)) with the rise of GA/EA concentrations and time duration during the period from 1 to 50 min and then remained constant from 50 min to 90 min. The molecular chain was stretched with charge repulsion and the capacity for swelling increased. Collagen hydrogels are sensitive to pH, indicating that ionic groups in hydrogels play an important role in absorbing water in the gel, as stated by Wang et al. [32]. This effect suggested that the GA/EA cross-linked hydrogels were suitable for application in acidic environments, where they could make good use of their absorptive property.

### 3.4. Thermogravimetric Analysis (TGA) of Collagen Hydrogels

Figure 3A,B, and Table 1 depict the thermal stability analysis of collagen hydrogels cross-linked with GA and EA. The thermal degradation characteristics of the hydrogels exhibited variability contingent upon the origin and concentration of phenolic compounds within GA or EA. The quantity of hydroxyl groups present in phenolic compounds is known to exert a substantial influence on the interaction between proteins and phenolic compounds [35,36]. The presence of carboxylic and hydroxyl groups in phenolic compounds could result in intra- and intermolecular interactions such as hydrogen, ionic, covalent, and non-covalent bonds, which change the chemical connotation of proteins and phenolic chemicals [35,37]. The interactions contributed to the improvement in thermal stability. In the thermogravimetric (TG) analysis, weight loss was observed in the initial stage (Δw1 = 6.5%, 99.8 °C) for all hydrogel samples. The weight loss during the specific temperature range was attributed to the release of two types of water states, namely, free and bound water, absorbed by the hydrogels [38]. Furthermore, the hydrogels treated with GA/EA lost less weight than the control group, implying that the presence of GA/EA reduced the water content of the collagen hydrogels due to the greater hydrophobicity of the phenolic compounds.

The second stage of weight loss was observed in all hydrogels at a temperature range of 228.3 °C–257.9 °C, with a weight loss percentage (Δw2) ranging from 2.6% to 9.7%. The weight loss at the second stage was typically associated with the release of structurally bound water and low molecular weight proteins, specifically collagen [38]. In conclusion, the thermal stability analysis reveals that the thermal degradation behavior of collagen hydrogels was influenced by the presence of phenolic compounds from GA/EA cross-linking. The interactions between proteins and phenolic compounds, driven by the hydroxyl and carboxylic groups, contributed to the increased thermal stability observed in the GA/EA cross-linked collagen hydrogels. The weight loss observed during TG analysis corresponds to the release of water states and low molecular weight proteins present in the collagen hydrogels [39]. The observed temperature range in the second stage of weight loss was found to be higher than the decomposition temperature of phenolic compounds [40]. The present study suggested that the cross-linking between collagen and GA/EA in the hydrogels occurs through hydrogen bonding. The lower weight loss (Δw2) observed in the GA/EA-treated collagen hydrogels compared to the control further supports the strong cross-linking between the protein (collagen) and phenolic compounds (GA/EA) via hydrogen or covalent bonds [37]. The decomposition temperatures (Td1 and Td2) of collagen protein-based hydrogels containing GA and EA were reduced compared to those of the control. On the other hand, the decomposition temperature changed with the different concentrations of GA/EA. The results indicated that hydrogels immersed in GA or EA solutions exhibited higher heat resistance than those of the control group.

During the third stage, which was characterized by Δw3 ranging from 25.7% to 52.4% and Td3 between 330.1 °C and 415.8 °C, the weight loss was attributed to the degradation of larger cross-linked proteins within the collagen hydrogels. Additionally, a fourth stage of weight loss (Δw4 = 7.1–19.1%) was observed in the temperature range of 489.3 °C to 541.3 °C. The weight loss was predominantly associated with the degradation of high-temperature stable components. The immersion of collagen hydrogels in GA or EA solutions at different concentrations significantly influenced the thermal stability of the hydrogels (*p* < 0.05), primarily through strong interactions between the proteins and phenolic compounds, and particularly through covalent cross-linking [37]. Notably, hydrogels immersed in 30% and 50% GA/EA concentrations, based on the dry weight of collagen, exhibited enhanced thermal stability as compared to all other hydrogels.

### 3.5. Differential Scanning Calorimetry (DSC) Measurements

DSC measurements of GA/EA cross-linked hydrogels are presented in Figure 3C,D. The collagen hydrogels immersed in GA or EA phenolic acids showed an improved Td value compared to the original hydrogels, as described in Table 2. When the collagen-based hydrogels were heated, the helix–coil conversion took place, and, as a result, the helix disappeared and was progressively separated into three randomly coiled peptide α-chains [41]. Typically, the thermal transformation of collagen has been proposed to be the collapse of the triple helical structure of collagen into random coils, with Td being the major endothermic peak [21,41]. The process of triple helical structure modification of GA/EA cross-linked collagen hydrogels was represented by a typical endothermic peak in the range of 56.78 °C to 75.78 °C. It has been found that when the structural integrity of collagen is improved, the endothermic peak is shifted to a lower temperature, resulting in a significantly lower Td than that of collagen with structural integrity [4,21,42]. The denaturation temperature (Td) increased to approximately 19 °C with the increasing concentration of GA or EA from 1 to 100% according to the dry weight of collagen. Moreover, GA 100% and EA 100% obtained the highest denaturation temperatures (Td) of 74.96 °C and 75.78 °C, respectively. However, the higher denaturation temperature (Td) observed in hydrogels cross-linked with GA or EA as compared to that of native collagen could be attributed to numerous factors. First, the presence of these cross-linking agents increases the thermal stability of the collagen hydrogels, as evidenced by the increase in Td with increasing concentrations of GA or EA. A possible explanation for the increase in thermal stability may be an increase in the accessibility of active sites in the collagen molecules [29]. The presence of GA or EA might be facilitated by the collagen interaction with cross-linkers, and the bonding between collagen and GA/EA could increase the resistance of the network to thermal denaturation and contribute to a more stable network structure [43]. The hydrophobic interfaces, mainly maintained by glycine residues and H-bonds formed between GA or EA depositions, play an important role in the stabilization of collagen molecules. In addition, the other possible reason could be that the GA/EA reactive sites could react with the amino group (NH_2_) of the collagen side chains [44]. In short, it was also noted that the thermal transition temperature represents the energy required for the destruction of the amide bonds, hydrogen bonds, and van der Waals forces that maintain the collagen triple helical structure. It should be noted that the increase in denaturation temperature is promising for maintaining the structural reliability of collagen, which is fundamental for the assembly of collagen-based hydrogels. Indeed, the increased hydrophobicity of GA or EA molecules, as well as their ability to form peptide bonds, can be important in their incorporation into certain regions of collagen fibrils [43]. This inclusion helps to stabilize the collagen scaffold structure. Because GA and EA are hydrophobic, they can interact with hydrophobic areas inside collagen fibrils. Hydrophobic interactions could happen between nonpolar areas of GA or EA and specific amino acid residues within collagen molecules, including proline or hydrophobic clusters. The interaction between hydrophobic binding improves GA or EA affinity and binding to collagen [45].

### 3.6. Analysis by X-Ray Diffraction (XRD)

The structural stability of collagen molecules after modification with GA or EA was evaluated using X-ray diffraction (XRD). Figure 4A,B presented the XRD spectra of GA/EA cross-linked collagen hydrogels. The XRD spectra revealed a diffuse scattering pattern with a small peak observed at about 8°, indicating the presence of intermolecular packing gaps between the molecular chains. An earlier study by Hu et al. [46], suggested that the observation was compatible with the Schmitt model’s description of the assembly of collagen molecules into fibrils [46]. Furthermore, a broader peak in the 20° to 25° range was detected, corresponding to the widespread scattering of collagen fiber synthesis [45]. The peak pattern in the XRD spectra of the GA/EA cross-linked hydrogels was identical to that of the control collagen hydrogels. The closeness implies that the collagen’s structural reliability was intact after treatment with GA or EA solutions. According to the XRD examination, the cross-linking of collagen with GA or EA had no significant effect on the overall structural properties of the collagen fibers. However, a previous study suggested that the presence of GA or EA in collagen hydrogels represented the intermolecular binding sites and that the assembly of collagen molecules into fibrils remained constant [47]. Therefore, the findings supported the assumption that GA/EA cross-linking does not alter collagen’s structural stability, making it an appropriate modification approach for retaining collagen’s inherent structure for biological applications [46,48].

### 3.7. Fourier Transform Infrared Spectroscopy (FTIR) Analysis

Fourier transform infrared spectroscopy (FTIR) was used to describe the chemical structure of collagen and collagen GA or EA cross-linked hydrogels. Figure 5A,B explain the amide bands of collagen fibers with or without GA or EA cross-linking. The amide A band, observed in the wave number range of 3430–3470 cm^−1^, was a characteristic feature of collagen cross-linked hydrogels. The amide A band was associated with hydrogen bonding interactions and results from the unfolding vibration of N-H bonds. Moreover, it provided valuable evidence about the network of hydrogen bonds within the structure of collagen. In addition, another band was detected in the wave number range of 3230–3290 cm^−1^, and was known as the amide B band. The asymmetric stretching of the CH_2_ groups within the collagen molecules was responsible for the amide B band. The conformational arrangement and structural properties of the collagen hydrogels can be determined from the presence and characteristics of the band. In addition, a band at 3140–3180 cm^−1^ was also observed in GA or EA cross-linked collagen fibers. However, the amide A and amide B bands vanished in pristine collagen hydrogels. In addition to the classic amide A and amide B bands, a novel peak in the wave number range of 1700–1730 cm^−1^ was identified in the spectra of collagen hydrogels cross-linked with GA or EA, but it was not present in the spectra of pristine collagen hydrogels. The unfolding vibrations of C=O (carbonyl) groups of chemical intermediates produced during GA or EA cross-linking are primarily responsible for this new peak. The emergence of the peak demonstrated that C=O groups participate in the cross-linking event, resulting in the production of more stable C-OH (hydroxyl) groups [45]. The unfolding vibrations of the C=O groups can provide insight into the structural changes and chemical modifications that occur within the collagen matrix throughout the cross-linking process. The inclusion of GA or EA could be the reason for these changes and could increase the stability and structural integrity of collagen-based hydrogels. Amides (A, B) promote consistency in the triple helical structure of collagen [49]. Amide (I) was located between 1620–1700 cm^−1^ and consisted of three main components: a band at 1650 cm^−1^ correlated with the α-helix/random coil confirmation, a band developed at 1620–1640 cm^−1^ corresponding to the β-sheet conformation, and a band appearing at 1660–1670 cm^−1^ corresponding to the β-turn [48]. In general, amide (I) was associated with the secondary structure of the protein and represents the trembling of amide carbonyls along with the polypeptide backbone and the native triple helix, which were transformed into C- and N- telopeptides in the collagen [49]. In addition, an amide (II) band of collagen fibers was found between 1550–1570 cm^−1^ and attributed to CH_2_ bending vibration. The C-N unfolding and N-H bending vibrations of the amidic bond, and the wiggle vibrations of the glycine backbone and the CH_2_ group of the proline side chain, are responsible for the complex amidic peaks [21]. On the other hand, FTIR shows strong binding as the concentration of GA or EA increased from 1 to 200%, except for GA 1% and EA 100% based on the dry weight of collagen, which might be due to the difference in their structure and binding ability at low and higher concentrations. The FTIR results indicated that GA or EA cross-linking has a significant effect on the structural properties and could be suitable for future experiments compared to the native collagen hydrogels, as the obtained results showed that clearer (amide A, B) and amide (I, II, III) bands at 1410–1480 cm^−1^, corresponding to the unfolding vibrations of CH_2_ and C=O, were altered in comparison to the native collagen hydrogels [50].

### 3.8. Dynamic Rheological Measurements

The storage modulus (G′) represents the elastic response of the hydrogels, and it was an indication of the ability to store and recover energy upon deformation. Moreover, the storage modulus (G′) also indicates the stiffness and rigidity of the hydrogel network. In contrast, the loss modulus (G″) reflects the viscous behavior and energy dissipation within the gel. The loss modulus (G″) was also known to measure the hydrogel’s ability to flow and deform under applied stress. The relationship between G′ and G″ aids in the understanding of the dominant behavior, i.e., whether the hydrogel was more elastic or more viscous. The loss factor, often expressed as the phase angle tangent (tan δ), was calculated as the ratio of G″; to G′. However, tangent (tan δ) provided valuable information about the relative contribution of the elasticity and the viscosity within the hydrogel. On the other hand, a low value of tan δ mostly indicates the elastic response where the storage modulus leads toward the loss modulus. Conversely, a higher value of tan δ indicates more viscous behavior; however, the loss modulus dominates. The frequency dependence curves of storage modulus *G*′, loss modulus *G*″, and tan δ of all COL-GA/EA hydrogels are presented in Figure 6A–F. The results obtained in the present study demonstrated that both the loss modulus (G″) and the storage modulus (G′) exhibit an increasing trend with increasing sweep frequencies up to 100% of GA or EA concentration based on the dry weight of collagen. However, both moduli decreased at 200% of GA or EA concentration. It was noteworthy that the storage modulus of elasticity (G′) was significantly higher than the loss modulus of elasticity (G′) at a constant frequency. The results indicated an expansion in the flexibility and a reduction in the mobility of the hydrogels after cross-linking with GA or EA [51,52]. The loss tangent (tan δ), which was the ratio of G″ to G′, serves as a measure of the transition from liquid-like to solid-like behavior. A value of tan δ close to 1 indicates a distinct elastomeric behavior [21]. In our study, all hydrogels exhibited tan δ values less than 1, signifying a superiority of solid-like gel properties. Moreover, tan δ showed a weak frequency dependence, suggesting the presence of stable gel-like properties in the GA/EA cross-linked hydrogels. In addition, tan δ showed a similar trend concerning the concentration of GA or EA, and was highest at a 100% concentration, and then decreased to the lowest value at a concentration of 200%, based on the dry weight of collagen. These findings suggested that the hydrogels maintain inter- and intramolecular cross-linked bonds between collagen and GA or EA components. There was evidence of improvement in the density of intermolecular bonds in collagen, which could be due to hydrogen bonding or hydrophobic interactions.

### 3.9. Enzymatic Degradation of Hydrogels

In the present study, the amount of hydroxyproline after enzymatic degradation was used to evaluate the enzymatic inhibition ability of pristine hydrogels and GA/EA cross-linked collagen hydrogels. Figure 7 shows the percentage residue after enzymatic degradation of pristine and cross-linked hydrogels after immersion in collagenase solution at 37 °C for 24 h. As the concentration of GA or EA in the collagen hydrogels was increased, the percentage of the residual amount of collagen was also increased. In addition, pristine hydrogels have significantly (*p* < 0.05) lower % residues as compared to GA/EA cross-linked hydrogels. The findings indicated that the resistance to collagenase of GA or EA cross-linked collagen hydrogels was significantly (*p* < 0.05) higher compared to that of the native collagen hydrogels. On the other hand, hydrogels cross-linked with GA have notably better resistance compared to EA, particularly at GA 30% to GA 100%; for example, GA 5%–100% achieved (28.25%, 42.09%, 56.20%, 62.41%, and 76.56%, respectively), and GA 200% achieved 30.50%. However, the EA percent (%) residue was enhanced from EA 5% to EA 30%, (9.08–47.66%), and then decreased with the increasing concentration of EA; for example, EA 50% reached 34.98, EA 100% reached 32.66, and EA 200% reached 29.86 percent residue. Furthermore, the results showed that the pristine collagen hydrogels have lower resistance to the collagenase enzyme compared to cross-linked hydrogels. Additionally, phenolic acid cross-linked hydrogels were primarily dependent on the formation of a covalent bond between the polymer and the cross-linking agent, e.g., via the Michael or Schiff reaction, and, as a result, such hydrogels had resistance to enzymatic degradation. Moreover, according to Bam et al. [43] GA is a mixture of -OH and -COOH groups and was described as a potential cross-linker by increasing collagen stability, as indicated by greater resistance to collagenase activity. These functional groups can react with functional groups found in collagen, such as the -NH_2_ and -OH groups. In general, the breaking of covalent bonds between intermolecular interactions, such as the van der Waals force and the hydrogen bond, requires more energy and has been associated with a lower degree of degradation [29].

### 3.10. Scanning Electron Microscopy (SEM)

The influence of GA or EA on the microstructure of collagen-based hydrogels was studied using scanning electron microscopy (SEM). Figure 8A–G demonstrates the SEM images and the surface areas of the hydrogels. SEM analysis revealed the presence of a typical tough assembly in both collagen and collagen-cross-linked hydrogels, which was attributed to the association of collagen molecules within the hydrogel matrix. In accordance with the results of XRD, FTIR, and physicochemical parameter analysis, the microstructure of the hydrogels showed changes depending on the presence of GA or EA and the concentration used. It was observed that hydrogels with a relatively rougher surface, showing an accumulation of material on the surface, were obtained at higher concentrations of GA or EA. As reported in previous studies [49,53], the increased irregularity of the hydrogel surface can be attributed to enhanced covalent and non-covalent interactions between collagen proteins and phenolic compounds [43]. The use of GA or EA to cross-link collagen molecules provides additional bonding opportunities, both between collagen molecules and within individual collagen molecules [33]. As a result, the structural properties of the hydrogel are densified and compacted. The modification in structural properties has been linked to the formation of intermolecular bonds between collagen molecules, resulting in a network of interconnected fibrils within the hydrogel matrix [34]. The hydrogels exhibited significant changes in microstructure as the concentration of GA or EA increased from 30% to 100% (as shown in Figure 8). In particular, the presence of fibrils within the hydrogels became more pronounced, and the number of fibrils increased at higher concentrations of either GA or EA. The hydrogels’ fibrillar structure led to their increased water-retaining capacity. The increased number of fibrils increased the surface area of the hydrogel and improved its ability to retain water molecules, resulting in higher swelling values when compared to pristine hydrogels. The creation of fibrils can be linked to an increase in the density of intermolecular interactions between collagen and GA or EA, which promotes the tangling and aggregation of collagen molecules, resulting in a more interconnected and porous structure. These intermolecular associations include covalent bonds, hydrogen bonds, and numerous non-covalent interactions. The increasing predominance of these intermolecular linkages causes the collagen molecules to become entangled and intertwined, resulting in a denser structure [53]. Additionally, intermolecular associations included covalent bonds, hydrogen bonds, and numerous non-covalent interactions. The increasing predominance of these intermolecular linkages causes the collagen molecules to become entangled and intertwined, resulting in a denser structure [54]. The formation of intramolecular associations inhibited the mobility of collagen molecules and the resulting hydrogels had a more compact and dense microstructure.

## 4. Conclusions

This study successfully presented fabricated collagen-based hydrogels cross-linked with GA or EA and demonstrated remarkable physicochemical, thermal, structural, and morphological properties. Cross-linking collagen-based hydrogels with GA and EA resulted in considerable improvements in a range of physicochemical parameters. These enhancements covered all the properties, such as the storage modulus, loss modulus, resistance to enzymatic degradation by type I collagenase, heat stability, porosity, swelling response, and water retention. The GA/EA cross-linking method was crucial in retaining the structural integrity of native collagen, as evidenced by XRD and FTIR spectroscopic investigations. Furthermore, SEM analysis demonstrated that the surface morphology of the GA/EA cross-linked hydrogels was improved when compared to the native hydrogels. Collectively, the results indicated that the incorporation of GA and EA into collagen hydrogels resulted in significant improvements in their enzymatic, thermal, and structural properties. These improvements were attributed to the formation of inter- and intramolecular cross-links between the collagen and GA/EA components. The cross-linked structure resulted in improved functional properties of the hydrogels by providing increased stability and resistance to degradation. The results of the present study indicated that the obtained GA/EA cross-linked collagen hydrogels are important for the design of collagen biomaterials for different biomedical applications and provide a promising approach to improve the properties and performance of collagen hydrogels for future studies.

## Figures and Tables

**Figure 1 polymers-15-04647-f001:**
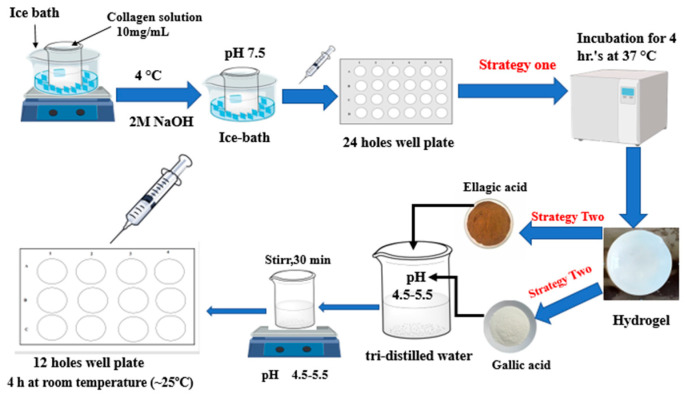
Schematic diagram demonstrating the overall strategy of the current work.

**Figure 2 polymers-15-04647-f002:**
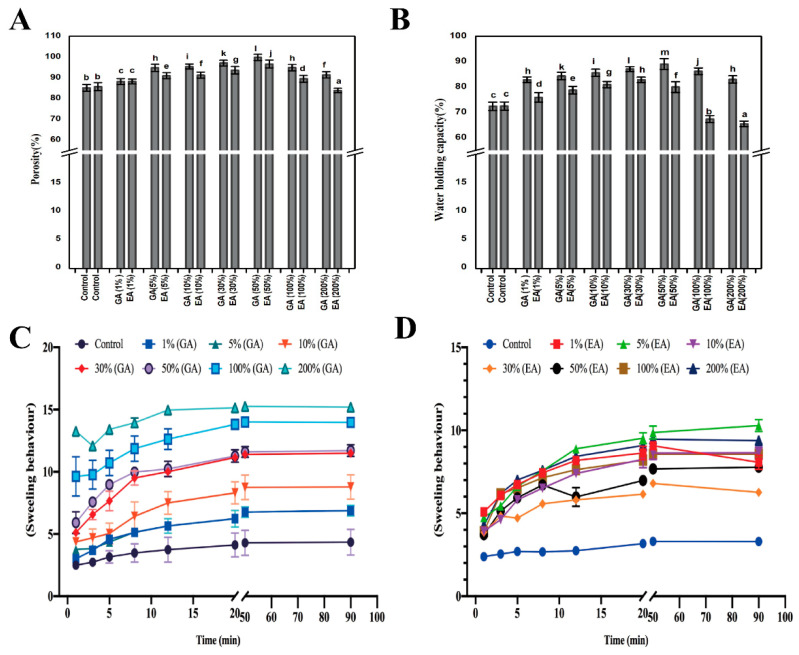
Evaluation of physical properties of pristine and GA/EA cross-linked hydrogels. (**A**) Porosity. (**B**) Water-holding capacity (WHC). (**C**,**D**) Swelling behavior of pure, GA, and EA hydrogels with different concentrations of gallic and ellagic acid according to the dry weight of collagen. Different letters (a, b, c, etc.) above concentrations indicate significant (*p* < 0.05) differences.

**Figure 3 polymers-15-04647-f003:**
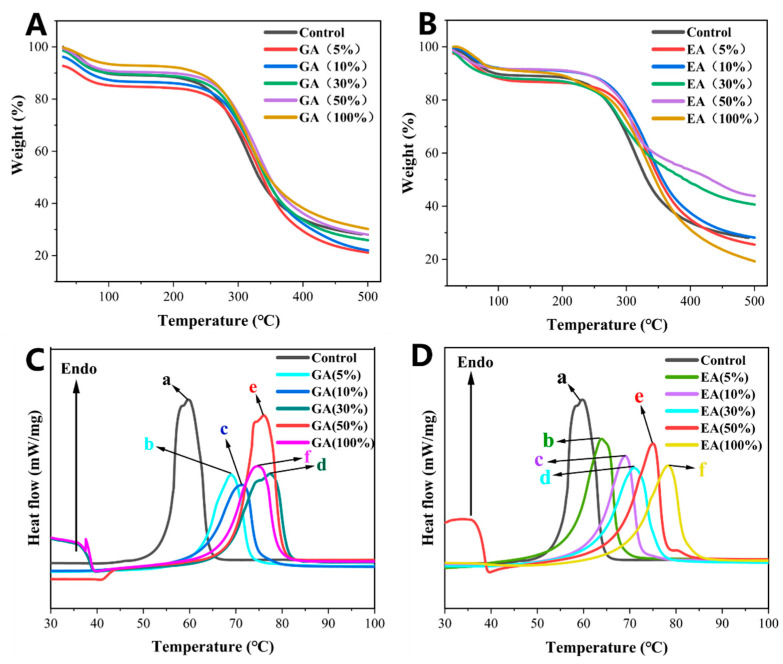
(**A**) TGA curves of GA cross-linked collagen hydrogels. (**B**) TGA curves of EA cross-linked collagen hydrogels. (**C**) DSC curves of GA cross-linked collagen hydrogels. (**D**) DSC curves of EA cross-linked collagen hydrogels. Different letters (a, b, c, etc.) above the concentrations indicate significant (*p* < 0.05) differences.

**Figure 4 polymers-15-04647-f004:**
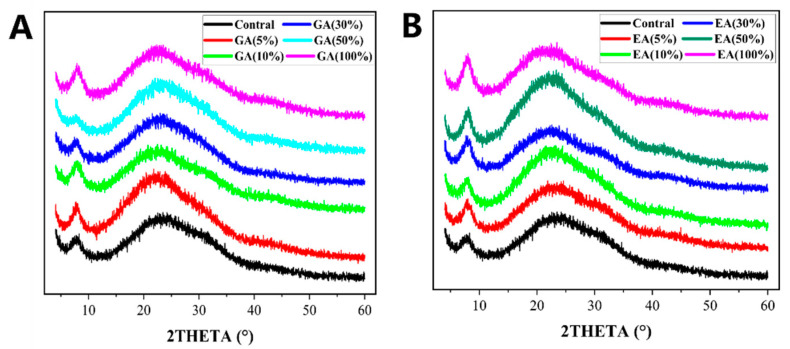
(**A**) XRD curves of GA cross-linked and pristine hydrogels with different concentrations of gallic acid according to the dry weight of collagen. (**B**) XRD curves of EA cross-linked and pristine hydrogels with different concentrations of ellagic acid according to the dry weight of collagen.

**Figure 5 polymers-15-04647-f005:**
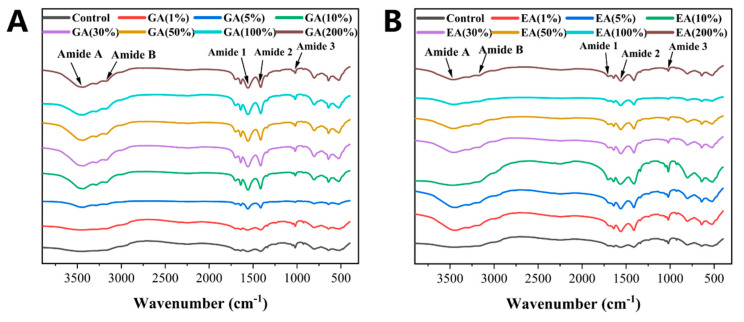
(**A**,**B**) FTIR spectra of GA or EA cross-linked or pristine collagen hydrogels with different concentrations of GA/EA according to the dry weight of collagen.

**Figure 6 polymers-15-04647-f006:**
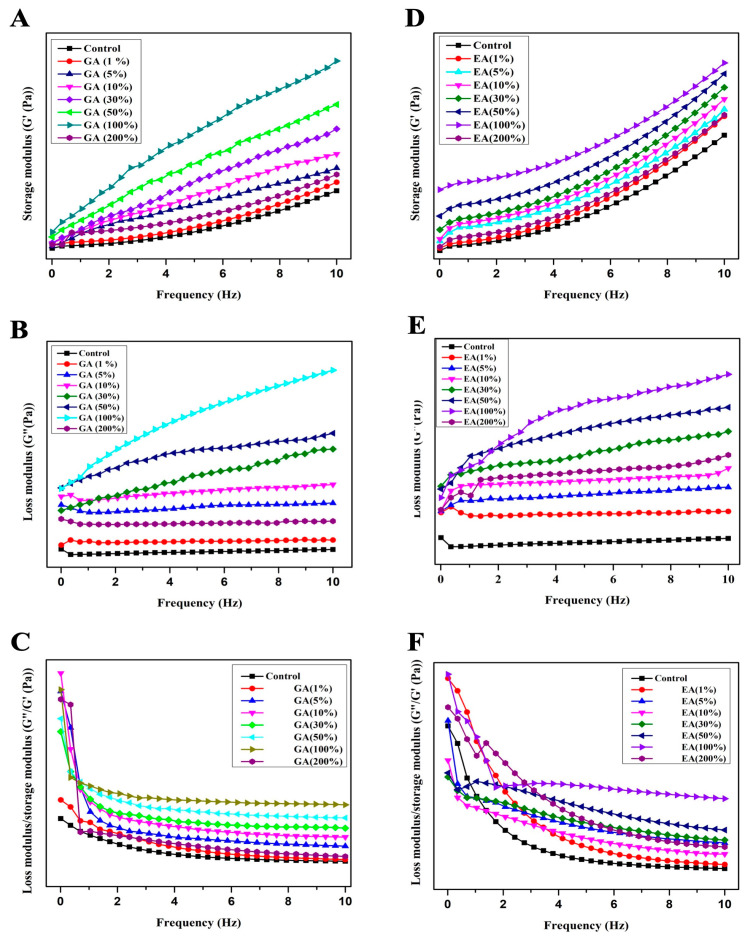
Effects of gallic acid and ellagic acid according to the dry weight of collagen with different concentrations on the dynamic viscoelasticity of GA/EA cross-linked and pure collagen hydrogels. (**A**) Storage modulus of gallic acid (G′). (**B**) Loss modulus of gallic acid (G″). (**C**) Loss factor of gallic acid (tan δ) (**D**) Storage modulus of ellagic acid (G′). (**E**) Loss modulus of ellagic acid (G″) (**F**) Loss factor of ellagic acid (tan δ).

**Figure 7 polymers-15-04647-f007:**
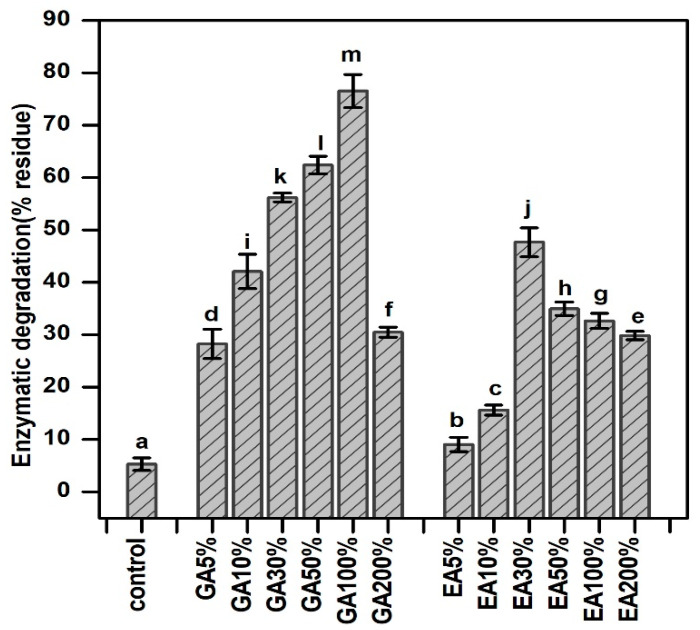
Enzymatic degradation of GA & EA cross-linked collagen hydrogels with different concentrations of gallic and ellagic acid (C) control, and 5%, 10%, 30%, 50%, 100%, and 200% were different concentrations according to the dry weight of collagen. Different letters (a, b, c, etc.) above the concentrations indicate significant (*p* < 0.05) differences.

**Figure 8 polymers-15-04647-f008:**
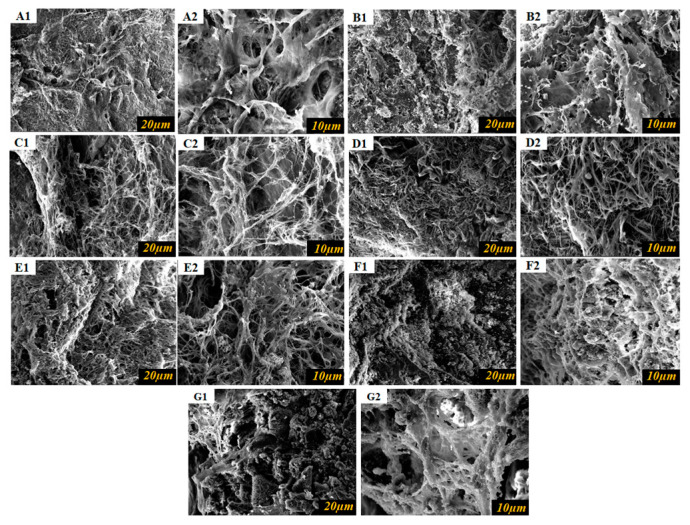
Morphological images of collagen hydrogels visualized by scanning electron microscopy. (**A1**,**A2**) SEM images of the pristine collagen; (**B1**,**B2**) SEM images of GA 30%; (**C1**,**C2**) SEM images of GA 50%; (**D1**,**D2**) SEM images of GA 100%; (**E1**,**E2**) SEM images of EA 30%; (**F1**,**F2**) SEM images of EA 50%; (**G1**,**G2**) SEM images of EA 100% crosslinked collagen hydrogels.

**Table 1 polymers-15-04647-t001:** Thermal degradation temperature (Td, °C) and weight loss (Δw, %) of collagen hydrogels immersed with GA and EA with different concentrations according to the dry weight of collagen. Δ_1_, Δ_2_, Δ_3_, and Δ_4_ represent the first, second, third, and fourth stages of weight loss of hydrogels during the heating scan, respectively.

Samples	(*w*/*w* of Protein)	Δ1		Δ2		Δ3		Δ4		Residue (%)
		Td1	Δw1	Td2	Δw2	Td3	Δw3	Td4	Δw4	
GA	Control	96.4	10.1	228.3	3.0	366.7	47.6	489.3	11.1	28.2
	5%	85.3	10.8	247.3	5.1	392.6	43.1	539.6	11.3	28.7
	10%	87.6	7.5	257.9	6.2	397.4	37.7	538.2	16.1	32.5
	30%	91.1	9.4	252.2	5.9	397.7	25.7	540.4	18.5	40.5
	50%	99.8	7.2	253.5	5.7	388.5	35.2	541.2	8.1	43.8
	100%	99.8	6.5	246.5	5.3	400.1	51.7	543.5	19.1	17.4
EA	5%	99.3	11.5	255.1	4.7	380.8	45.6	540.3	14.3	23.9
	10%	91.4	6.5	251.3	5.3	406.1	50.4	540.1	10.7	27.1
	30%	77.5	11	234.1	2.6	415.8	40.2	541.3	7.1	39.1
	50%	89.7	7.3	232.5	3.3	330.1	27.9	539.8	18.6	42.9
	100%	84.3	7.9	256.1	9.7	405.4	52.4	540.6	13.9	16.1

**Table 2 polymers-15-04647-t002:** Thermal denaturation (Td, °C) of collagen hydrogels immersed with GA and EA with different concentrations according to the dry weight of collagen.

Treatments	Sample	Thermal Denaturation Temperature (Td) °C
GA	Control	56.78
	5%	63.35
	10%	65.88
	30%	68.93
	50%	71.19
	100%	74.96
EA	5%	62.15
	10%	66.13
	30%	67.84
	50%	70.51
	100%	75.78

## Data Availability

The data presented in the current study is available on request from the corresponding author.

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
