# Peer review of "Effects of Phenolics on the Physicochemical and Structural Properties of Collagen Hydrogel"

_polymers, 2023, doi:10.3390/polym15244647_

Round 1

Reviewer 1 Report

Comments and Suggestions for Authors

Manuscript namely “Development of collagen hydrogels with improved physico-chemical properties modified by natural phenolics” studied the impact of different phenolics on the collagen hydrogels. Authors have conducted several analyses to support their objectives. This paper could be useful for the scientific community as well as the medical industry. However, the quality of writing is poor. The result and discussion need to be improved. Authors are giving only one explanation “due the difference in chemical properties and nature of compounds” for all the analyses.  ****Moreover, English grammar must be edited by the native speaker as authors have used present tense along with the past tenses.

**** All the statements made by the authors are not supported by the appropriate references. In addition, comparison must be compared with the other studies.

Hence, this MS must be rewritten, discussion should be strengthened, and grammar needs to be improved before submission.

Detailed comments:

Title is not informative, not all the concentrations of phenolics improve the hydrogels. Authors can change accordingly ‘effects of phenolics on the physicochemical and structural properties of collagen hydrogel” or “Fabrication of collagen hydrogel as affected by the gallic and ellagic acids”. Authors can amend as per the suggestions.

Abstract:

Some of the numerical results should be presented in the abstract.

L15: What functional properties were studied; reviewer cannot see in the methodology.

Introduction

L38: Please check reference style, is it [1-3] or [2], [1], [3]. See the author’s guidelines.

L38-41 is unclear and confusing. Please revise it for better understanding.

Objectives are also confusing and some sentences are repeated (L76-80 and 82-84). Also check the font size, it is not same as other text.

Materials and method

Some methods need to be written clearly; it is not sure what is happening there. What kind of samples were used in each method? How those samples were prepared must be clarified. For example, for porosity or WHC and rheology the same types of samples can not be used.

L89 and 97: Scientific name must be in italics.

L100: How isopropanol can adjust the pH?

L103: What is 1:3000?

L105: (NH4)2SO4>>> (NH4)2SO4

L105: For purification, 1.5 M (NH4)2SO4 was added overnight, followed by 0.5 M acetic acid. Did author added (NH4)2SO4 overnight? Or stirred overnight?

-Purified collagen was resolubilized in 0.5M acetic acid, dialyzed with 0.04M and 0.02M disodium dihydrogen pyrophosphate solutions, and finally double-distilled water for 2, 3, and two days, respectively.>>>Please revised the sentence it is not clear, whether the mixed solution used or separate solvents were used for dialysis.

-2, 3, and two days, need to be consistent.

L112: Pure collagen: What was the purity?

Phenolics were used up to 200%, why such concentration was used? There is a huge gap that 1% and 200%.  It simply means that hydrogels have more of phenolic compounds rather than the collagen. Gel structure will deteriorate if the concentration of native gelling agent is reduced. This is mainly due the dilution effect.

L188: what was the size of wells?

In gel formation, authors first gelatinized the collagen hydrogel (L118-119) and then added crosslinkers (L121 and 125)? How can those compounds interact with collagen molecules once it is solidified?

L126: Authors did wash after the treatment, then it is obvious that some of the phenolics were washed away? So it might not be the exact content as mention by the authors (1-200%)

This section (2.3) must be revised for the better understanding. It is not clear how the gels were made.

IN section 2.4: It is not clear ether the authors used fresh hydrogel or freeze dried or soaked gel samples. Moreover, no information was given in the section, whether the samples were dried or not. This must be clarified.

If authors have freeze dried the gel prepared in section 2.3, then it must be notify.

For example: L131: lyophilised sample (when did this prepare?), L132 degassed permeate the hydrogel (not clear) and L139 freeze-dried cylindrical specimen as well as L206, where freshly prepared samples were used?

L134: How scrapping was done, as this method can affects the final results.

L134: What is the meaning of left-half?

Format of section 2.4.3 heading and 2.4.4 are not consistent. Please check in all text.

For rheology, how the samples were prepared? Authors can not use the samples that are already solidified (hydrogel). Please provide the detailed as it is not clear :self-assembled neutral collagen?

L190: What is enzyme hydrolysate?

L191: Sigma, city and country name must be given. Moreover, I suggested to move this information in the material and chemical section.

L201: Gallic acid and ellagic acid: authors are switching between the full name and abbreviation. Please be consistent with it. (L231-232, etc.) check it in whole text.

L210L Freeze-drier’s name is given here. Was it different than those used in previous section. If not, then no need to repeat again.

Results and discussion

Results from each sample must be explained and discussed here. No reason was given between the different concentrations as well as between different phenolics. Authors need to explain the reason at least in the first results. For example: L228, at higher content of GA/EA, porosity decreased but why?

Citation must be give when making statements.

L224-226: Not clear, rephrased it.

L236: Col-GA/Col-EA: please explain this term and use this term throughout the MS or no need to use it at all.  

Figure 1: Please compare the EA and GA samples (Provide statistical analysis between them). Authors no need to add Control bar alone with the each concentration. It can mislead the readers. Please remove it.

-Authors have used Col-GA/Col-EA codes for the samples. If you still need to use it, please provide in the figures also. Otherwise use same the name in whole text.

Figure caption: (A, B, C, D) Evaluation of physical properties of COL-GA & EA hydrogels. A) Porosity B) Water holding capacity (WHC). C, D) Swelling behavior of COL-GA & EA hydrogels with different concentrations of gallic and ellagic acid according to the dry weight of collagen. Different letters (a, b, c etc.) above concentrations indicate significant (p < 0.05) difference.

Change to

Figure 1. Evaluation of physical properties of COL-GA & EA hydrogels. A) Porosity B) Water holding capacity (WHC). C, D) Swelling behavior of COL-GA & EA hydrogels with different concentrations of gallic and ellagic acid according to the dry weight of collagen. Different letters (a, b, c etc.) above concentrations indicate significant (p < 0.05) difference.

Add A or B in front of the sample or analysis            , do same with caption Fig 3 and Fig 4, check for others too.

L253: concentration>>>bars.

L244: is >>> was or were, this must be checked in whole text. Author have made such mistake most often.

Section 3.2 water holding capacity >>>>Water holding capacity

See the WHC in L256 and L257. Once WHC had been already explained, no need to repeat again and again.

L261: is >>> was …

L272: This difference can be due to GA and EA's differing chemical characteristics and cross-linking capacities: how do they have different capacities. Please elaborate.

L327: Elaborate and support with previous study.

Figure 2 C and D: what are those letters (a-d). Rewrite the caption, captions should stand alone.

Table 1: Add stat, SD and units of values?

L344: and low molecular weight proteins present in the collagen hydrogels. How did they form?

L347: support with appropriate reference.

L313-330 are repeated version of L339-365.

L387: NH2. Please check

DSC results seems like review of literature. No results were explained here. The data in table 2, is not reported anywhere.

L473-474: Provide citation same for the L475-476.

L492 and 479 are same.

Citation for L572-528 and 528-531.

In Fugure 6: where is the letter a on the bars. Compare GA and EA samples. Authors need to explained why 100% samples have the highest degradation.

Where is the 200% sample?

Figure 7. SEM images are poor. These images should support the porosity of the gel samples. However, nothing is visible in here.

Comments on the Quality of English Language

English grammar must be improve. Several mistakes.

Author Response

Thank you for your valuable suggestions

Reviewer 2 Report

Comments and Suggestions for Authors

This study by Munir et al, is a well-designed and executed study that investigates the effect of cross-linking collagen hydrogels with gallic acid (GA) and ellagic acid (EA) on their physicochemical properties. The authors used a variety of techniques to characterize the hydrogels, including XRD, FTIR, SEM, and swelling tests. The results of the study showed that cross-linking with GA and EA significantly improved the storage modulus, thermal stability, swelling behavior, porosity, and resistance to type 1 collagenase of the hydrogels. The authors suggest that the improvement in properties is due to the formation of hydrogen bonds between the phenolic hydroxyl groups of GA and EA and the nitrogen atoms of the collagen backbone. They also suggest that potential hydrogen bonding or hydrophobic interactions may be involved. Overall, I think this is a well-done study that provides valuable insights into the interaction between GA/EA and collagen. The results suggest that GA and EA can be used to improve the physicochemical properties of collagen hydrogels, making them a potential material for use in biomedical applications. The study is interesting and can be published after addressing the following points:

1.     The authors suggest in line 20 that the improvement of collagen properties is due to the formation of inter- or intramolecular bonds within the hydrogel structure, as indicated by the SEM results. However, it is important to note that SEM is a technique to study the morphology of materials, and it cannot confirm bonding or chemical interactions.

2.     In Figure 4 of the FTIR, more than one spectrum were merged, which decreased the quality of the peaks. I suggest using only three spectra in each figure: the control, GA or EA at 50% concentration, and GA or EA at 100% concentration. The rest of the spectra can be transferred to the supporting information.

3.     It is preferable to draw an illustration scheme of the method of gel preparation.

4.     The authors of the study claim that the modification of collagen with GA and EA does not affect the structure of the collagen based on the XRD analysis. However, this is an illogical conclusion. XRD is a technique that is used to study the structure of inorganic materials and to study the crystallization in organic or inorganic materials. The XRD spectra of the collagen hydrogels cross-linked with GA and EA show a clear effect on the crystallization of collagen. Therefore, the authors should re-interpret the results and calculate the crystallinity percentage.

Comments on the Quality of English Language

Language should be revised and typos corrected

Author Response

                                     Comments and Suggestions for Authors

                                             Response to Reviewer 2

Title: Development of collagen hydrogels with improved physicochemical properties modified by natural phenolics.

This study by Munir et al, is a well-designed and executed study that investigates the effect of cross-linking collagen hydrogels with gallic acid (GA) and ellagic acid (EA) on their physicochemical properties. The authors used a variety of techniques to characterize the hydrogels, including XRD, FTIR, SEM, and swelling tests. The results of the study showed that cross-linking with GA and EA significantly improved the storage modulus, thermal stability, swelling behavior, porosity, and resistance to type 1 collagenase of the hydrogels. The authors suggest that the improvement in properties is due to the formation of hydrogen bonds between the phenolic hydroxyl groups of GA and EA and the nitrogen atoms of the collagen backbone. They also suggest that potential hydrogen bonding or hydrophobic interactions may be involved. Overall, I think this is a well-done study that provides valuable insights into the interaction between GA/EA and collagen. The results suggest that GA and EA can be used to improve the physicochemical properties of collagen hydrogels, making them a potential material for use in biomedical applications. The study is interesting and can be published after addressing the following points:

First of all, on behalf of all authors, I would like to express my sincere appreciation for the administrative constructive comments, considerable time and significant efforts on our manuscript. I am again thankful to you for valuable suggestions on our manuscript Draft ID: polymers-2577595 entitled ‘Development of collagen hydrogels with improved physicochemical properties modified by natural phenolics’. Those comments and suggestions are very valuable and helpful for revising and improving our manuscript, response is as follows:

Question 1

The authors suggest in line 20 that the improvement of collagen properties is due to the formation of inter- or intramolecular bonds within the hydrogel structure, as indicated by the SEM results. However, it is important to note that SEM is a technique to study the morphology of materials, and it cannot confirm bonding or chemical interactions.

Response: Thank you. we have revised according to your suggestions in the revised manuscript.

Before Modifications

In the current era, collagen hydrogels treated with natural phenolics to enhance their physicochemical properties have received much attention. The aim of this research was to prepare collagen hydrogels cross-linked with gallic acid (GA) and ellagic acid (EA) at different concentrations depending on the collagen dry weight. The functional properties of the native collagen hydrogels were compared with those of the GA/EA crosslinked hydrogels. XRD and FTIR spectroscopic analyses confirmed the structural stability and reliability of the collagen after treatment with either GA or EA. Cross-linking also significantly contributed to the improvement of storage modulus, thermal stability, swelling behavior, porosity, and resistance to type 1 collagenase, especially in the hydrogels immersed in 30%, 50%, and 100% concentrations of GA/EA. In addition, SEM analysis revealed the formation of inter- or intramolecular bonds within the hydrogel structure, resulting in a denser and more compact architecture in the GA/EA cross-linked hydrogels. The formation of hydrogen bonds between the phenolic hydroxyl groups of GA and EA and the nitrogen atoms of the collagen backbone is likely responsible for the improved physicochemical properties. Potential hydrogen bonding or hydrophobic interactions are suggested by the presence of inter- and intramolecular cross-links between collagen and GA or EA components, along with an increased density of intermolecular bonds. Overall, this study paves the way for further investigations in this field by providing valuable insights into the interaction between GA/EA and collagen.

After Modifications:

     In the current era, the treatment of collagen hydrogels with natural phenolics for the improvement of physicochemical properties has been the subject of considerable attention. The aim of the research was to prepare collagen hydrogels cross-linked with gallic acid (GA) and ellagic acid (EA) at different concentrations depending on the collagen dry weight. The structural, enzymatic, thermal, morphological and physical properties of the native collagen hydrogels were compared with those of the GA/EA cross-linked hydrogels. XRD and FTIR spectroscopy analyses confirmed the structural stability and reliability of the collagen after treatment with either GA or EA. The cross-linking also significantly contributed to the improvement of storage modulus, 435 Pa for 100% GA cross-linked hydrogels. The thermal stability was improved, as 43.8% highest residue weight was obtained at 50% GA cross-linked hydrogels compared to all other hydrogels. The hydrogels immersed in 30%, 50% and 100% concentrations of GA also showed improved swelling behavior, porosity and the highest resistance to type 1 collagenase 76.56% was obtained at 50% GA cross-linked collagen hydrogels. Moreover, GA 100% and EA100% obtained the highest denaturation temperature (Td) 74.96°C and 75.78°C respectively. In addition, the SEM analysis was also carried out to check the surface morphology of the pristine collagen hydrogels and the cross-linked collagen hydrogels. The result showed that the hydrogels cross-linked with GA/EA were denser and more compact. However, the improved physicochemical properties were probably due to the formation of hydrogen bonds between the phenolic hydroxyl groups of GA and EA and the nitrogen atoms of the collagen backbone. The presence of inter- and intramolecular cross-links between collagen and GA or EA components, along with an increased density of intermolecular bonds, suggests potential hydrogen bonding or hydrophobic interactions. Overall, present study paves the way for further investigations in the field by providing valuable insights into the GA/EA interaction with collagen molecules.

Question 2

In Figure 4 of the FTIR, more than one spectrum was merged, which decreased the quality of the peaks. I suggest using only four spectra in each figure: the control, GA or EA at 30%, 50% concentration, and GA or EA at 100% concentration. The rest of the spectra can be transferred to the supporting information.

Response: Thank you, we have revised according to suggestion in the revised manuscript.

Before Modifications

Figure 4. (A, B). FTIR spectra of GA or EA cross-linked or pristine collagen hydrogels with different concentrations of GA/EA according to the dry weight of collagen.

After Modifications:

Figure 4. (A, B). FTIR spectra of GA or EA cross-linked or pristine collagen hydrogels with different concentrations of GA/EA according to the dry weight of collagen.

Question 3

It is preferable to draw an illustration scheme of the method of gel preparation.

Response: Thank you, we have revised according to suggestion in the revised manuscript.

Modified:

Question 4

The authors of the study claim that the modification of collagen with GA and EA does not affect the structure of the collagen based on the XRD analysis. However, this is an illogical conclusion. XRD is a technique that is used to study the structure of inorganic materials and to study the crystallization in organic or inorganic materials. The XRD spectra of the collagen hydrogels cross-linked with GA and EA show a clear effect on the crystallization of collagen. Therefore, the authors should re-interpret the results and calculate the crystallinity percentage.

Response:  I am grateful for the chance to clarify the concerns raised about our study's conclusion that collagen modification with gallic acid (GA) and ellagic acid (EA) did not affect collagen structure as determined by XRD analysis. Your assessment has prompted us to rethink our findings thoroughly, and I want to react logically and scientifically.

  You accurately point out that XRD is commonly used to investigate the crystalline structure of materials, most of which are inorganic, or to investigate the crystallization behavior of both organic and inorganic substances. We acknowledge that XRD is a valuable method for studying structural changes, including crystalline alterations. However, clarification of the precise aims of our XRD analysis and its relevance to our study is required.

The overall structural integrity of the collagen-based hydrogels was the primary focus of the XRD analysis in our study, particularly in the context of the natural triple helix structure of collagen.  Conversely, you are correct in stating that XRD is commonly utilized for the research of we wanted to see if the insertion of GA and EA as cross-linkers caused any noticeable disruption or change of this triple helix conformation, which is important to collagen's biological activity. Our XRD study revealed no substantial changes in the triple helix structure of collagen after cross-linking with GA and EA. However, we completely agree with your finding that XRD can detect crystalline changes in collagen that are connected to its structural order.  

Round 2

Reviewer 1 Report

Comments and Suggestions for Authors

The manuscript (MS) has been improved significantly. Authors should carefully examine the MS to make it flawless. After some changes, MS is in publishable form.
Some of the minor changes should be focused.
L56: gallic acid to GA.  L72-78: ellagic acid to EA
as those names had already been abbreviated. Check for such content in all MS.
L102: Define CDU.
Check L105: To neutralize the pH, skin pieces were immersed in 10% (w/w) isopropanol for 24 hours after being neutralized with double-distilled water. Please change as mentioned in the cover letter.
I think the authors forgot to remove this section (105-114). DO CHECK IN ALL MS.
L110: correct (NH4)2SO4
Please keep consistency in some words such as hrs or hours or h.
L303: water-holding capacity (WHC) and the again water-holding capacity (WHC) in line 309.

Author Response

                                Comments and Suggestions for Authors

                                              Response to Reviewer 1

Title: Effects of phenolics on the physicochemical and structural properties of collagen hydrogel

The manuscript (MS) has been improved significantly. Authors should carefully examine the MS to make it flawless. After some changes, MS is in publishable form.

First of all, on behalf of all the authors, I would like to express my sincere appreciation for the constructive comments of the administrative staff, the considerable amount of time, and the considerable efforts spent on our manuscript. Once again, I want to thank you and the reviewers for the valuable suggestions in our manuscript, Draft ID: polymers-2577595, titled "Effects of phenolics on the physicochemical and structural properties of collagen hydrogel". The following is our response to these comments and suggestions, which are very valuable and helpful in the revision and improvement of our manuscript:

Question 1

Some of the minor changes should be focused on, like L56: gallic acid to GA.  L72-78: ellagic acid to EA, as those names had already been abbreviated. Check for such content in all MS.

Response: I would like to express my sincere appreciation for the comments and the considerable amount of time and effort you have spent on our manuscript.

Before Modifications

Plant-based phenolic composites are considered to be the most vital bioactive composites. These compounds contain a number of hydroxyl groups and have diverse biological functions such as structural support, pigmentation, chemical defense, and radiation prevention [7, 8]. Gallic acid (GA) was also a phenolic compound of plant origin and is known as (3,4,5-trihydroxybenzoic acid). Moreover, gallic acid has been shown to have a variety of properties in biomedical studies, including anti-allergic, anti-fungal, anti-inflammatory, anti-cancer, anti-viral, anti-mutagenic, and anti-carcinogenic properties [7, 9, 10, 11]. In addition, from a medical point of view, gallic acid plays an important role in the protective mechanism against reactive oxygen species and free radicals. It breaks the free radical chains through hydroxyl groups [6, 12]. Furthermore, gallic acid as a bioactive and stable agent in chitosan/PVA-based hydrogels was reported Thanyacharoen et al.[13]. It was also reported by Jiang et al.[14] that gallic acid can increase the release efficiency of chitin-based hydrogels.

On the other hand, ellagic acid (EA), a representative of flavonoids found in a variety of fruits such as pomegranate, pecans, and berries, etc., has received extensive attention due to its numerous antioxidant, cytotoxic, radical scavenging ability, , antiviral, anti-inflammatory, anticarcinogenic and anti-apoptotic properties [15, 16]. Ellagic acid is a dilactone of hexahydrooxydiphenic acid. It is usually produced by plants through the hydrolysis of tannins such as ellagitannins [17]. It contains four hydroxyl groups and these hydroxyl groups may increase the Antioxidative action of lipid peroxidation to protect the cell against oxidative destruction [15].Therefore, ellagic acid has hydrophilic characteristics due to its structural appearance, which includes a planar biphenyl, a lipophilic component connected by two lactone rings and four hydroxyl groups. These hydroxyl groups combine with the lactone groups to form a hydrophilic unit [18]. The hydrophilic region of the ellagic acid molecule plays an important role in its biological activity. Notably, it is due to the presence of both hydrogen-bonding acceptor (lactone) and donor (-OH) sites. In particular, the phenolic hydroxyl groups in ellagic acid can separate under physiological conditions [19, 20].On the other hand, it has been reported that ellagic acid can cross-link with PEG-based hydrogels and reduce the viability of human oral cancer cells by Huang et al.[21].

The study could contribute to provide a new vision in the biomaterials industry or the biomedical industry by the cross-linking of collagen hydrogels with natural phenolic compounds such as gallic acid (GA) or ellagic acid (EA).

For the preparation of crosslinked collagen hydrogels, GA and ellagic acid (EA) were dissolved individually in tri-distilled water at different concentrations (0%, 1%, 5%, 10%, 30%, 50%, 100%, and 200% w/w based on the dry weight of the collagen hydrogels) in tri-distilled water at a pH range of 4.5-5.5.

Collagen hydrogels cross-linked with gallic acid/ellagic acid for morphological characterization were prepared as described by Liu et al.[23] with slight modifications.

Moreover, gallic acid according to Bam et al.[46], is a mixture of -OH and -COOH groups and described as a potential cross-linker by increasing collagen stability as indicated by greater resistance to collagenase activity.

Study successfully presented fabricated collagen-based hydrogels cross-linked with gallic acid (GA) or ellagic acid (EA), and demonstrated towards remarkable physico-chemical, thermal, structural, and morphological properties.

Specifically, in the absence of gallic acid or ellagic acid, the percentage of hydroxyproline released from the collagen-based hydrogel was compared to that of fully degraded collagen

After Modifications

Plant-based phenolic composites are considered to be the most vital bioactive composites. These compounds contain a number of hydroxyl groups and have diverse biological functions such as structural support, pigmentation, chemical defense, and radiation prevention [7, 8]. Gallic acid (GA) was also a phenolic compound of plant origin and is known as (3,4,5-trihydroxybenzoic acid). Moreover, GA has been shown to have a variety of properties in biomedical studies, including anti-allergic, anti-fungal, anti-inflammatory, anti-cancer, anti-viral, anti-mutagenic, and anti-carcinogenic properties [7, 9, 10, 11]. In addition, from a medical point of view, gallic acid plays an important role in the protective mechanism against reactive oxygen species and free radicals. It breaks the free radical chains through hydroxyl groups [6, 12]. Furthermore, GA as a bioactive and stable agent in chitosan/PVA-based hydrogels was reported Thanyacharoen et al.[13]. It was also reported by Jiang et al.[14] that GA can increase the release efficiency of chitin-based hydrogels.

On the other hand, ellagic acid (EA), a representative of flavonoids found in a variety of fruits such as pomegranate, pecans, and berries, etc., has received extensive attention due to its numerous antioxidant, cytotoxic, radical scavenging ability, , antiviral, anti-inflammatory, anticarcinogenic and anti-apoptotic properties [15, 16]. Ellagic acid is a dilactone of hexahydrooxydiphenic acid. It is usually produced by plants through the hydrolysis of tannins such as ellagitannins [17]. It contains four hydroxyl groups and these hydroxyl groups may increase the Antioxidative action of lipid peroxidation to protect the cell against oxidative destruction [15].Therefore, EA has hydrophilic characteristics due to its structural appearance, which includes a planar biphenyl, a lipophilic component connected by two lactone rings and four hydroxyl groups. These hydroxyl groups combine with the lactone groups to form a hydrophilic unit [18]. The hydrophilic region of the EA molecule plays an important role in its biological activity. Notably, it is due to the presence of both hydrogen-bonding acceptor (lactone) and donor (-OH) sites. In particular, the phenolic hydroxyl groups in EA can separate under physiological conditions [19, 20].On the other hand, it has been reported that EA can cross-link with PEG-based hydrogels and reduce the viability of human oral cancer cells by Huang et al.[21].

The study could contribute to provide a new vision in the biomaterials industry or the biomedical industry by the cross-linking of collagen hydrogels with natural phenolic compounds such as GA or EA.

For the preparation of crosslinked collagen hydrogels, GA and EA were dissolved individually in tri-distilled water at different concentrations (0%, 1%, 5%, 10%, 30%, 50%, 100%, and 200% w/w based on the dry weight of the collagen hydrogels) in tri-distilled water at a pH range of 4.5-5.5.

Collagen hydrogels cross-linked with GA/EA acid for morphological characterization were prepared as described by Liu et al.[23] with slight modifications.

 Moreover, GA, according to Bam et al.[46], is a mixture of -OH and -COOH groups and described as a potential cross-linker by increasing collagen stability as indicated by greater resistance to collagenase activity.

Study successfully presented fabricated collagen-based hydrogels cross-linked with GA or EA, and demonstrated towards remarkable physico-chemical, thermal, structural, and morphological properties.

Specifically, in the absence of GA or EA, the percentage of hydroxyproline released from the collagen-based hydrogel was compared to that of fully degraded collagen

Question 2

L102: Define CDU

Respond: Thank for you for your valuable comment. It has been answered according to your suggestion.

Answer

CDU = collagen digestion unit

One collagen digestion unit (CDU) releases peptides from bovine Achilles tendon collagen equivalent in ninhydrin color to 1.0 mole of leucine in 5 hours at pH 7.4 at 37°C in the presence of calcium ions. At 25°C, one FALGPA hydrolytic unit hydrolyzes 1.0 moles of furylacryloyl-leu-gly-pro-al at a rate of 1.0 moles per minute. A neutral protease unit hydrolyzes casein to produce a color equivalent to 1.0 mole of tyrosine per 5 hours at pH 7.5 and 37°C. In the presence of DTT, one clostripain unit will hydrolyze 1.0 moles of BAEE per minute at pH 7.6 and 25°C.

Question 3

Check L105: To neutralize the pH, skin pieces were immersed in 10% (w/w) isopropanol for 24 hours after being neutralized with double-distilled water. Please change as mentioned in the cover letter.

Respond: Thank for you for your valuable comment. It has been improved according to your suggestion.

Before Modifications

To neutralize the pH, skin pieces were immersed in 10% (w/w) isopropanol for 24 hours after being neutralized with double-distilled water.

After Modifications

The skin pieces were then immersed in 10% (w/w) isopropanol for 24 hours to remove fat then neutralized with double-distilled water.

Question 4

I think the authors forgot to remove this section (105-114).

Response: Thank you, we have deleted according to suggestion in the revised manuscript

Question 5

DO CHECK IN ALL MS.L110: correct (NH4)2SO4

Response: Thank you, we have revised according to suggestion in the revised manuscript

Before Modifications

For purification, 1.5 M (NH4)2SO4 was added overnight, followed by 0.5 M acetic acid.

 After Modifications

 For purification, 1.5 M (NH4)2SO4 was added overnight, followed by 0.5 M acetic acid.

Question 6

Please keep consistency in some words such as hrs or hours or h.

Respond: Thank for you for your valuable comment. It has been improved according to your suggestion.

Before Modifications

Dehydration of the frozen hydrogels was performed in a lyophilizer for minimum 24 hrs.

After Modifications

Dehydration of the frozen hydrogels was performed in a lyophilizer for minimum 24 hours.

Question 7

L303: water-holding capacity (WHC) and the again water-holding capacity (WHC) in line 309.

Response: Thank you, we have revised according to suggestion in the revised manuscript.

Before Modifications

Conversely, the water-holding capacity (WHC) of collagen hydrogels crosslinked with EA increased with increasing EA content, up to a 30% concentration before declining gradually. The highest WHC was found in collagen-based gallic acid hydrogels at 50% GA concentration (89.07%), whereas the lowest WHC was found in collagen-based ellagic acid hydrogels at 200% EA concentration (64.92%). These findings highlight the significant influence of GA and EA cross-linking on WHC, which may have effects for a variety of biomaterial applications. Lin et al.[31], conducted research that supports the idea that increased WHC in hydrogels was beneficial for tissue regeneration and tissue engineering by permitting superior nutrient retention inside the hydrogel matrix.

After Modifications

Conversely, the water-holding capacity (WHC) of collagen hydrogels crosslinked with EA increased with increasing EA content, up to a 30% concentration before declining gradually. The highest WHC was found in GA cross-linked collagen hydrogels especially at 50% GA concentration (89.07%), whereas the lowest WHC was found in EA cross-linked collagen hydrogels at 200% EA concentration (64.92%). These findings highlight the significant influence of GA and EA cross-linking on WHC, which may have effects for a variety of biomaterial applications. Lin et al.[31], conducted research that supports the idea that increased WHC in hydrogels was beneficial for tissue regeneration and tissue engineering by permitting superior nutrient retention inside the hydrogel matrix.
